# On the Historical Background and Ideological Resources of the Confluence of Islam and Confucianism

**Wei Wang**

College of Philosophy, Nankai University, Tianjin 300350, China; wangwei0175@nankai.edu.cn

**Abstract:** From the Yuan to the mid-Ming period, the people of Huihui (回回人) in mainland China gradually Sinicized in terms of their languages, family names, marriages, costumes, and ethical values. There was close interaction between these Muslims and Confucian scholars in China. Most of the mosque inscriptions in this period were written by Confucian scholars, who were the first to try to interpret Islam in Confucian terms. Around the mid-Ming period, the Chinese language became the lingua franca of Muslims in mainland China, and the teaching of Arabic and Persian classics in Chinese became an urgent need at this time. It was at this time that the Confucian academies were revived with the government's permission. Thereupon, the Muslim scholar Hu Dengzhou (胡登洲) founded a rejuvenated educational system known as Jingtang education (經堂教育), which produced a group of Muslim scholars who wrote in Chinese. Islam thus entered the historical arena of interaction with traditional Chinese religions. During the middle and late Ming period, changes in political and economic structures led to changes in the general mood of society. The rise of Wang Yangming's Mind Study (心學) brought a lively academic atmosphere and a relaxed cultural environment to intellectual circles. The concept of "The same mind and the same principle of the sages in the East and the West" advocated by Lu Jiuyuan (陸九淵) and Wang Yangming (王陽明) was taken seriously by Muslim scholars and became a crucial theoretical reference in their writing process. In the late Ming and early Qing periods, the classical learning of the Shandong school and the Jinling school of Jingtang education focused on the study of Xingli (性理學). The theory of Sufism shared many common ideas with the Three Teachings (Confucianism, Taoism, and Buddhism) which showed a tendency towards confluence in the Song, Yuan, and Ming periods. Chinese Muslim scholars, known as Huiru (回儒), drew intellectual resources from all of these traditions to construct their study of Xingli.

**Keywords:** Islam; Confucianism; inter-civilizational dialogue; historical background

## 1. Introduction

As one of the best stories in the history of human civilizational exchange, the confluence of Islam and Confucianism is generally regarded as a scholarly activity initiated by Chinese-speaking Muslim scholars in the Ming and Qing periods, who creatively interpreted Islamic thought in Confucian terms and promoted a profound Islamic–Confucian dialogue. To date, a large amount of relevant research has been published, including the collection, collation, and translation of these scholars' works, the examination of these books and their references, the interpretation of these texts, and the study of their ideas. A small number of scholars have also focused on what motivated these Muslim scholars to write. These views will be responded to in the conclusions of this paper. In short, most of these studies focus on the writing activities of Muslim scholars in late Ming and early Qing China, emphasizing the characteristics of these texts and the ideas contained within them.

This study is based on the position that the emergence of any kind of thought or cultural phenomenon is closely related to a specific social context and is inevitably influenced to some extent by the social groups, cultural institutions, historical situations, and spirit of the times. Thinking is never a special activity that can be free from the influence

of group life. Throughout the history of Islam, its spread and development have been inseparable from specific social and cultural contexts, so Islam in any period and region must be understood in its own social and cultural context. Muslims around the world are ethnically, linguistically, socially, and culturally diverse, embodying both a unified and diverse Islamic civilization. Although they all believe in Islam, they do not all understand and practice Islam in the same way.

This study is concerned with a broader sense of the confluence of Islam and Confucianism, which is not limited to the scholarly activities of Muslim scholars in the late Ming and early Qing periods who interpreted Islam in Confucian terms. In fact, theories of Islam, Confucianism, Buddhism, and Taoism were creatively integrated into a new kind of philosophical thought. It is important to emphasize that this intellectual activity is inextricably linked to the historical and socio-cultural contexts of the Yuan, Ming, and Qing periods. An exploration of these contexts will help us to better understand this unique cultural phenomenon and to thus explore ways to live in harmony among civilizations and uphold the beauty of each civilization and the diversity of civilizations around the world, which has unique value in contemporary times.

## 2. The Sinicization of the People of Huihui and Changes in the Understanding of Islam by Confucian Scholars since the Yuan Period

Islam was introduced to China in the seventh century. Muslims who came to China through the "Silk Road" and the "Maritime Silk Road" were mainly merchants, envoys, and travelers. Official contacts between the Tang Dynasty and the Arab Empire began in the second year of Yong Hui (永徽二年, 651 A.D.), during the reign of the caliph Othman (Li et al. 2007, p. 29). In the subsequent hundred years, there were 39 official exchanges recorded in Chinese historical books. In the Tang period, foreign Muslims lived in Xi'an, Yangzhou, Guangzhou, Quanzhou, Hainan, Sichuan, Yunnan, and other places. In the Song period, mosques and Muslim cemeteries were built in Guangzhou, Quanzhou, Mingzhou (Ningbo), Hangzhou, Yangzhou, Beijing, Xi'an, Kaifeng, Zhengzhou, and other places. Due to the implementation of the square market (坊市) system at that time, these Muslims, called Fanke (蕃客), could only live in demarcated areas, named Fanfang (蕃坊), in these cities. During the Tang and Song periods, Chinese people had limited knowledge of Islam and there were many misrepresentations or misunderstandings. They generally believed that Muslims worshipped heaven, that Muhammad was similar to Buddha, and that Muslim rituals were exotic laws.

At the beginning of the 13th century, the western invasion of the Mongolian army altered this situation as the people in Central Asia and West Asia were forcibly involved in the Mongols' conquest of the Southern Song Dynasty, at which point a large number of Muslims entered China. In addition, trade between the new Yuan Dynasty and other Asian countries resulted in a large number of Muslims visiting China to conduct business, who largely lived in big cities and coastal areas. In the Yuan Empire, Muslims known as the Huihui belonged to a group called the Semu people (色目人) and "were in all places in the Yuan Dynasty" (Zhang 1974, p. 8598). Their status was second only to that of the Mongols, and many Muslim officials were relied upon by the rulers. During the period of Mongol rule, most of the Muslims entering China came from the eastern Islamic world, which was under the influence of Persian culture and where "Persian language had dominated in place of Arabic language since the 10th century" (Liu 2013, p. 142). By the time of the Mongol invasion in the 13th century, Islamic civilization under the influence of Persian culture throughout Persia and Central Asia was already highly developed. In Mongol-ruled China, "the Huihui people's cultural outlook shows a strong color of Persian civilization, so the language to communicate with the various ethnic groups within the Huihui people can only be Persian. Over time, the Persian language gradually replaced the original language of various Huihui people who entered China. It became the common communication language within them, as well as an important communication language for Huihui people to communicate with other social classes" (Liu 2013, p. 142).

During the Yuan period, Muslims maintained Islamic law in terms of the five pillars and halal diets, burials, and marriages, but they had to follow the state law in the public sphere, e.g., criminal cases, household registration, trade, and lawsuits (Li et al. 2007, pp. 204–16). During the reign of Yuan Emperor Renzong (元仁宗), the promotion of Sinology further accelerated the Sinicization of the Huihui people in terms of their languages, surnames, marriages, and costumes. In 1313, the emperor ordered the restoration of the imperial examination system (科舉) and made the *Commentaries on the Four Books* (四書集 註) the standard text for the examination. At this point, Zhu Xi's Neo-Confucianism (朱熹 理學) became the official school of the Yuan Dynasty, dominating state ideology. There were many Muslims who were educated in Confucianism and took the imperial examinations to become officials (Chen 2016, p. 140).

At the end of the Yuan Dynasty, some Hui Muslim generals made outstanding contributions to the peasant revolts against the Yuan and became some of the founding officials of the Ming Dynasty. In the Ming Dynasty, on the one hand, the rulers gave preferential treatment to Muslims, and many mosques were built at the emperor's command. On the other hand, some assimilation measures were taken against ethnic minorities, such as bans on Hu suits (胡服), Hu languages (胡語), and Hu surnames (胡姓), as well as a ban on maritime trade and a prohibition against same-ethnic marriage. The *Ming Law* (大 明律) states that "any Mongolian or Semu people are allowed to marry Chinese people, and they are not allowed to marry their own people. Those who violate the law will be flogged 80 times, their men and women will become rulers' slaves" (Liu 1998, p. 65). The ideological influence of Zhu Xi's philosophy reached its peak in the early Ming Dynasty when Zhu Di, the third emperor of the Ming Dynasty, ordered the compilation of three great books, namely the *Great Book of the Four Books* (四書大全), the *Great Book of the Five Classics* (五經大全), and the *Great Book of Xingli* (性理大全), which became the highest authority representing political ideology. Not only did Cheng and Zhu's Neo-Confucianism deeply influence Muslim intellectuals who read these books, but its ethical values also gradually penetrated the Chinese Muslim population through the institutions of clans and the local regulations of townships. All of this influenced the thinking of Chinese Muslims.

Following the Yuan period, Confucian intellectuals knew more about Islam than their predecessors. There were many Huihui Muslims who were officials or businessmen in the Yuan Dynasty, and they interacted frequently with Confucian intellectuals to improve their understanding. Before the mid-Ming period, the Chinese inscriptions in mosques were mainly written by Confucian scholars. It is noteworthy that these authors generally understood Islam in a Confucian way, believing that Islam was different from Buddhism and Taoism but very similar to the Tao of Confucius and Mencius (孔孟之道). We regard the inscriptions written by these Confucian scholars as the earliest documents that bridge Islam and Confucianism, and they represent the ideas of Confucian intellectuals. Today, we can find two inscriptions from the Yuan period and a number of inscriptions from the pre-Ming period. The earliest inscription that can be found today is *Inscription of Rebuild the Mosque* (重建禮拜寺記), written in 1348 by Yang Shouyi (楊受益), the magistrate of Anxi County (安喜縣尹), for the Dingzhou (定州) Mosque. The inscription reads:

> I think that Confucianism is the most popular religion in China, followed by Buddhism and Taoism, which both advocate nothingness and annihilation and are not free from delusion. Moreover, they abandon human morality, evaded taxes, and lead the people of the world to the point of disowning their fathers and monarchs, so what is there to say about their teachings? Only the religion of the Huihui is such a religion: their mosques do not contain idols, and are only empty buildings, because they follow the system that came from the Tianfang country in Western Regions . . . . . . One is not allowed to seek images of The Creator, and if an image of Him was produced, He would be similar to something, which is considered blasphemy against him; one can only express sincerity through thoughts, not idols. The beauty of Huihui tradition is known as well. Additionally, they obeyed the laws [of our country], undertook labor and paid taxes, so the righteousness of

ruler and minister [in their religion] is no different [from Confucianism]. Elders are loving, while youngsters are filial, so the affection of father and son is no different [from Confucianism]. And even with regard to the separate functions of husband and wife, the proper order of elders and juniors, and the faithfulness of friends, all of these are no different [from Confucianism]! Not only does their "no form, no image" coincide with the meaning of "silently and odorless" in the *Zhou Ya*, but their perfection of the Five Relationships also coincides with the meaning of the Five Rules in the *Zhou Shu*, without any difference. Is not their religion very different from Buddhism and Taoism? Moreover, they worship Heaven five times a day and fast once a year; they bathe no matter whether it is winter or summer; they give alms to relatives and strangers alike, which are signs of the sincerity of their faith and behavior, and there is nothing to criticize! 予惟天下之教儒教尚矣。下此而曰釋與老，虛無寂滅，不免於妄，且其去人倫、逃租賦，率天下之人而入於無父無君之域，則其教又何言哉。惟回回之為教也，寺無像設，惟一空殿，蓋祖西域天方國遺製。 ... ... 蓋造物主不可以形跡求，若擬之像則類物，殆亦瀆矣；惟有想無像，以表其誠，其遺風流俗之美，蓋可知也。況其奉正朔，躬庸租，君臣之義無所異；上而慈，下而孝，父子之親無所異；以至於夫婦之別，長幼之序，朋友之信，舉無所異乎！夫不惟無形無像，與周雅無聲無臭之旨吻合；抑且五倫全備，與周書五典五惇之義又符契，而無所殊焉。較之釋老不大有間乎！且其拜天之禮，一日五行；齋戒之事，每歲一舉欤夫！沐浴無間與寒暑，施與不問其親疏，則又其篤信力行而無所訾議焉者也! ([Yang 2012](), pp. 276–78)

As a Confucian, Yang describes Islam as being distinct from Buddhism and Taoism and in line with Confucianism in three respects: object of faith, ethics of behavior, and religious practice. First, he argues that Islam believes in a Creator who is invisible and transcendental, which is consistent with the idea expressed by "Heaven acts silently and odorless", a line from the *Book of Odes* (詩經). Second, he believes that the ethics of Islam are similar to the norms of the five relationships established by the Sage and King Shun (舜) in ancient times. As Mencius said, "According to the way of man, if they are well fed, warmly clothed, and comfortably lodged but without education, they will become almost like animals. The Sage (emperor Shun) worried about it and he appointed Hsieh to be minister of education and teach people human relations, that between father and son, there should be affection; between ruler and minister, there should be righteousness; between husband and wife, there should be attention to their separate functions; between old and young, there should be a proper order; and between friends, there should be faithfulness" ([Chan 1969](), pp. 69–70). The Five Relations have formed the general pattern of traditional Chinese society. Third, according to Yang, although Muslims' practices such as prayer, fasting, ablution, and Zakāt are different from the practices of Confucianism, they are also a manifestation of their commitment to faith, which is beyond criticism.

The second Yuan-period inscription is the *Inscription of Qingjing Mosque* (清淨寺記), written in 1350 by Wu Jian (吳鑒), a scholar from Sanshan (Fuzhou), for the Qingjing Mosque in Quanzhou (泉州). One important passage is worth noting:

In the beginning, Payghāmbar Muhammad, the king of Medina, was born with miracles and great virtues, and countries of the Western Region obeyed him and called him sage. The name Payghāmbar (پیغامبر) is equivalent to the Chinese word for the Messenger of Heaven, probably because people called him so out of respect. His teaching advocates that all things come from Heaven, and that Heaven is a Principle without any image. Therefore, they served Heaven with the utmost sincerity and did not placed any idols [in their mosques]. They fasted for one month every year, changed their clothes and bathed, and did not live in their daily dwellings. Every day, they faced the west and worshipped Heaven, and read the Scripture (Qur'an) with a pure heart. The Scripture was given by the heavenly beings (angel) and consisted of thirty collections, one hundred and fourteen parts (chapters), and six thousand six hundred and sixty-six volumes

(verses), with profound meaning. It establishes righteousness, justness, rectifying the mind and cultivating virtue as principles, and praying for the sovereign, educating the people, helping others with their difficulties and needs as tasks. Confessing faults, cultivating personal lives and do not causing trouble to others, being careful in internal and external edicts, all of these are not allowed the slightest violation of principle. 初，默德那國王別諳拔爾謨罕驀德生而神靈，有大德，臣服西域諸國，鹹稱聖人。別諳拔爾，猶華言天使，蓋尊而號之也。其教以萬物本乎天，天一理，無可像，故事天至虔，而無設像。每歲齋戒一月，更衣沐浴，居必易常處。日向西拜天，淨心誦經。經本天人所授，三十藏，計一百一十四部，凡六千六百六十六卷，旨義淵微。以至公無私、正心修德為本，以祝聖化民、周急解厄為事。慮悔過自新，持己接人，內外慎敕，不容毫末悖理。([Yang 2012](), pp. 257–58)

This text not only describes Allah as Heaven (天) or the Principle of Nature (理) in Confucianism but also mentions Islamic fasting, ablution, and prayer, as well as the Qur'an. It focuses on the content of Islam in human relations and moral teachings, such as "rectifying the mind and cultivating virtue", "praying for the sovereign and educating the people", "helping others with their difficulties and needs", and "cultivating personal lives and not causing trouble to others". These are clearly interpretations of Islam in the context of Confucianism. Islam was clearly presented in Confucian terms.

In the early Ming Dynasty, a large number of mosque inscriptions were written by Confucians, whose knowledge of Islam came from two sources: (1) from historical books and documents and (2) probably from brief introductions made by Muslim leaders in mosques. From the mid-Ming period onward, more and more Confucian intellectuals with a Muslim identity were also invited to write inscriptions, most of whom were government officials. Although they were Muslims, they grew up learning about Confucianism, took the imperial examinations, and had little knowledge of Islamic sciences.

The translation activities of Muslim scholars first began in the early Ming period. In the first year of the Ming Dynasty, the emperor Zhu Yuanzhang (朱元璋) set up an institution related to astronomy and calendars and summoned dozens of Huihui scholars to Nanjing to discuss the calendar. In 1382, the emperor summoned Muslim scholars and officials such as al-Haydarī (海答爾), Adā' al-Dīn (阿達兀丁), Mashā'ikh (馬沙亦黑), and Mahammad (馬哈麻). They were asked to translate the Arabic and Persian astronomical and calendrical books into Chinese ([al-Haydarī. 2008](), p. 5). In the preface of the extant *Book of Astronomy* (天文書), it is stated that:

The Principle of Heaven has no image, and He created mankind with infinite grace and generosity. Man's gratitude and thankfulness to Heaven are also endless. However, the great Dao in heaven and earth is inscrutable and unknowable. It is necessary for a wise sage to appear, whose heart is able to comprehend the subtleties of the Dao, and to establish the teaching in his time. After him, the worthies succeeded one after another. They also grasped the subtleties of the teachings of the ancient sages to teach them to the next generation. After the sage Muhammad, came generations of worthies, and those who made great contributions to [the preaching of] the Dao are clearly identifiable. 天理無象，其生人也，恩厚無窮。人之感恩而報天也,心亦罔極。然而大道在天地間，茫昧無聞，必有聰明睿智聖人者出，心得神會斯道之妙，立教於當世。後之賢人，接踵相承，又得上古聖人所傳之妙，以垂教於來世也。聖人馬合麻，及後賢輩出，有功於大道者，昭然可考。([al-Haydarī. 2008](), pp. 9–10)

The translation of the *Book of Astronomy* did not create a boom in the translation of Islamic texts among Chinese Muslims because most Muslim religious figures had a limited ability to express their religious ideas in Chinese. It should not be forgotten that before the formation of the Chinese-speaking Muslim ethnic community, the common language of the Muslims in China was Persian. The exploration of Islamic sciences in Chinese really began in the mid- and late Ming periods.

### 3. Chinese Muslims' Classical Learning Made a Breakthrough in the Context of Social and Cultural Changes in the Mid- and Late Ming Periods

Around the middle of the Ming Dynasty, a Chinese-speaking Muslim ethnic community had formed in mainland China. The Chinese language had become their native language, while Arabic and Persian had become foreign languages that they generally did not understand. How to use the Chinese language to teach Arabic and Persian Islamic books became an urgent problem. Most Muslim Zhangjiao (mosque leaders) and teachers knew Arabic and Persian, but they were not able to explain Islamic books and teachings well in Chinese, much less write in Chinese.

There were exceptions, of course, such as Chen Si (陳思), a mosque leader and scholar of the Jinan South Mosque, Shandong Province, who wrote the *Inscription of Origin and Return* (來復銘) in 1528. Chen was born to a family of Muslim scholars and grew up in the hometown of Confucius and Mencius, so he would have had a good understanding of Confucianism, which would have qualified him to write the inscription. Chen's *Inscription of Origin and Return* was the earliest text written by a Muslim scholar to introduce Islamic thought in Chinese. It is composed of one hundred and fifty-five Chinese characters and reads as follows:

> The Ultimate of Non-being and the Great Ultimate, the Two Modes and The Five Agents, originating from the Inaudible, began with the Invisible. The Emperor descends the law and gives us Mandate (spirit), which comes into existence with birth and forms with form. When Humanity is embodied in man's conduct, it is the Way. The principle and the apparatus are dependent on each other. Saints and fools have different talents, but what they are given is the same. Therefore, the mind is the citadel [of the nature], and the nature is the body [of the Dao]. From the Great Vacuity, there is Heaven. From the transformation of material force, there is the Way. In the unity of the Great Vacuity and material force, there is the nature (of man and things). Additionally, in the unity of the nature and consciousness, there is the mind. One should preserve his mind and nature in order to serve his Heaven. One must be careful to cultivate his personal life in order to await for the Mandate. One should Hold Fast to Seriousness and investigate principle to the utmost in order to nourish the nature. One should be cautious [over what he does not see] and apprehensive [over what he does not hear], in order to realize the Dao. One must feel no qualms, even in private places, in order to serve the heart. This is what it means to be friends with the Creator. Otherwise, what Heaven has bestowed is forsaken in vain by man, and how will he reply to God's command? 無極太極，兩儀五行，元於無聲，始於無形。皇降衷彝，錫命吾人，與生俱生，與形俱形。仁人合道，理器相成。聖愚異稟，予賦維均。是故心為郭廓，性為形體。繇太虛，有天之名；繇氣化，有道之名；合虛與氣，有性之名；合性與知覺，有心之名。存心與性，以事其天；慎修厥身，以俟此命；主敬窮理，以養此性；戒慎恐懼，以體此道；不愧屋漏，以事此心。斯與造物為徒矣。不爾，天顧畀之，人顧拚之，其將何以復帝者之命？

In this brief text, Chen quotes from the Chinese classics such as the *Book of Changes* (易經), the *Book of Odes* (詩經), the *Book of Mencius* (孟子), and the *Doctrine of the Mean* (中庸) and texts by other Confucian scholars such as Han Yu (韓愈), Zhou Dunyi (周敦頤), Shao Yong (邵庸), Zhang Zai (張載), and Zhu Xi (朱熹) to discuss Islamic thoughts on origin and return (Wang 2020).

Chen Si lived at a time when Chinese society was undergoing a series of changes. The social hierarchical structure formed in the early Ming Dynasty was markedly divided and consolidated, driving a transformation in people's social and economic relations. "The number of people who engaged in multiple occupations, commerce, industry and handicrafts increased, and so did the number of people who left the farmland. While it was generally believed that the social crisis of peasant disengagement from the land was caused by land annexation, heavy taxation, official corruption and usury exploitation, in fact, monetization of silver was undoubtedly one of the important factors contributing to

peasant disengagement from the land. Monetization greatly contributed to the increased commercialization of agriculture and the tendency of peasants to become non-peasants. Directly related to the growing number of people from all social classes involved in the silver monetary economy was the rise and growth of a professional merchant class. Merchants spread throughout the country in the late Ming period" (Wan 2011, p. 768). Beginning in the Jiajing (嘉靖) period, the phenomenon of abandoning agriculture and Confucianism for business became quite common, and the idea that "business is also the basis" began to emerge.

Influenced by economic changes, social ideology and traditional order were relaxed, and the dominance of Cheng–Zhu philosophy was challenged by various competing ideas. One of the most influential was Wang Yangming's Study of Mind. Cheng–Zhu's spirit of rational inquiry and genuine search for fundamental principles had, by Wang's time, degenerated into trifling with what Wang called "fragmentary and isolated details and broken pieces". Wang's "whole emphasis is on moral values. He was convinced that if the mind is divided or devoted to external things, it will be concerned only with fragmentary details and will lack the essentials. Scholars with such a mind will trifle with things and lose their purpose in life. For him this was the reason for the decline of the Confucian teachings, which in turn brought on the intellectual, political, and moral decay of his time" (Chan 1969, pp. 655–56). The prevalence of Wang Yangming's school among scholars brought a breeze of freedom to Chinese intellectual circles, breaking the unified ideology of Cheng and Zhu and giving the intellectual class space to speak.

In the mid-Ming period, academies were approved by the government due to the various drawbacks exposed by the Confucian official schools. Wang Yangming's thoughts and academic activities greatly promoted the flourishing of the academies, and the prosperity of the academies also provided the most suitable form and place for the dissemination of his doctrine. In 1509, Wang was invited to lecture at the Wenming Academy (文明書院) and advocated the doctrine of the unity of knowledge and action, bringing a new trend to intellectual circles. In 1524, he was invited to lecture at Jishan Academy (稷山書院), and in 1525 he founded Yangming Academy (陽明書院) in Shaoxing, where his disciples also began to lecture and spread Wang's thought. In 1528, he built Fuwen Academy (敷文書院) in Guangxi. Wang's learning became popular, and his disciples all over China built academies everywhere. These academies advocated opening up to the general public and educating the common people. The Taizhou school (泰州學派) is a typical example, which promoted Yangming's Study of Mind, opposed the fettering of human nature, advocated giving play to the nature of human beings, and was closer to the masses of the lower classes, leading the trend towards intellectual liberation in the late Ming period.

At that time, the religious education of Chinese Muslims was facing a situation of "the lack of scriptures, the lack of scholars. Neither the translation is clear, nor the explanation is groundless" (Yu and Lei 2001, p. 513). Affected by Confucian academies, Chinese Muslim scholars also started to set up private schools to allow more Muslims to gain knowledge of their religion and to train much-needed religious talents (Wang 2022). The most fruitful of these attempts was that of Hu Dengzhou (1522–1597), who established a rejuvenated educational system in Shaanxi. Hu was educated in Confucianism and Islamic sciences from an early age, and he always wanted to carry forward Islamic learning. During his business trip to Beijing at the age of fifty, he not only invited famous Confucian scholars to teach him the *Book of Odes* (詩), the *Book of History* (書), and Neo-Confucianism for further study, but he also studied with a foreign Muslim Shaykh for Islamic learning (Zhao 2022, p. 72). Hu explored a set of methods to explain and teach Islamic classics in Chinese in a way that was accurate, plain, and in line with Arabic grammar. After the establishment of Hu's educational system, namely Jingtang education, the teachings and ideas of Islam became more widely known to the Chinese people. Therefore, "it has truly become a religion with high cultural grade and complete posture among Chinese traditional religions" (Li 2013, p. 36). Islam stepped onto the historical stage of interaction with Chinese traditional religions.

Hu's educational system soon spread throughout the country, producing a number of Muslim scholars. By the early Qing period, Jingtang education had established several major hubs in China. Around the two largest centers, Jining (濟寧) and Nanjing (南京), the Shandong School (山東學派) and the Jinling School (金陵學派) were formed. Like Hu Dengzhou, many of the disciples in these two schools were proficient in Confucianism, and even Buddhism and Taoism. They rectified the early situation whereby Islamic scholars could not accurately express religious ideas in Chinese. These scholars (see Table 1), in a cultural context, dominated by Confucianism and the convergence of Confucianism, Taoism, and Buddhism, translated Islamic classics from Arabic and Persian into Chinese, or wrote with reference to these books, using words, concepts, and ideas of Chinese philosophy in their writing to affirm the commonality of Islamic and Chinese cultures in order to achieve better understanding and communication. Huiru (回儒), a term used to refer to these scholars who were both Muslims and Confucians, was coined by the Japanese scholar Kuwata Rokuro (桑田六郎) and has been adopted by today's scholars (Kuwata 1925).

Huiru generally recognized the concept of "the same mind and the same principle of the sages in the East and the West" (東海西海，心同理同), and this universalist view of truth, emphasized by the Study of Mind, provided a theoretical reference for their writing. As Lu Jiuyuan (陸九淵), the predecessor of Wang Yangming, said, "The four directions plus upward and downward constitute the spatial continuum (yü). What has gone by in the past and what is to come in the future constitute the temporal continuum (chou). The universe (these continua) is my mind, and my mind is the universe. Sages appeared tens of thousands of generations ago. They shared this mind; they shared this principle. Sages will appear tens of thousands of generations to come. They will share this mind; they will share this principle. Over the four seas sages appear. They share this mind; they share this principle" (Chan 1969, pp. 579–80). The influence of this idea was first reflected in a mosque inscription, probably written in the early 16th century, called "The Inscription on the Founding of the Mosque", which may have been written by someone in the Jiajing period of the Ming Dynasty in the name of Wang Gong (王珙), a Tang Dynasty official, which is now located in the Huajue Lane Mosque in Xi'an. The text reads:

> I have heard that what has lasted for hundreds of generations without perplexities is the Dao; what has lasted for hundreds of generations and still corresponds is the Mind. Only the Mind and the Dao of the sages are the same, that is, they corresponded each other and had no perplexities. Therefore, within the four seas (the entire world), there are sages. These so-called sages are those whose Minds and Way (Dao) are the same. The sage of the Western Regions, Muhammad, was born after Confucius and lived in the country of Tianfang. He was far away from the time and place of the Chinese sage. The language was not the same, but the Dao was the same. Why? Because the Minds of both were the same, so the Dao was also the same. In the old days, it was said that thousands of sages had the same Mind, tens of thousands of generations had the same Principle, and this is indeed true! 竊聞俟百世而不惑者，道也；曠百世而相感者，心也。惟聖人心一而道同，斯百世相感而不惑。是故四海之內，皆有聖人出。所謂聖人者，此心此道同也。西域聖人謨罕默德，生孔子之後，居天方之國，其去中國聖人之世之地，不知其幾也。譯語矛盾而道合符節者，何也？其心一，故道同也。昔人有言，千聖一心，萬古一理，信矣! (Yang 2012, p. 289)

The above statement that the sage of China and the sage of Islam had the same Mind and the same Dao was adopted by Liu Zhi, a famous Muslim scholar in the early Qing period, who also put forward the idea that "the teachings of the sages are the same in the East and the West, and the same in ancient and modern" (聖人之教，東西同，古今一) in his *Tianfang Dianli* (《天方典禮》). Perhaps, as William C. Chittick said, Huiru never lost sight of the fundamental goal of both traditions; that is, the transformation of the human self and the achievement of "one body with Heaven and earth" (Chittick 2011).

Thanks to the relatively free intellectual and cultural environment, Huiru in the late Ming and early Qing periods often responded to and criticized Confucianism, Buddhism, and Taoism on certain topics on which they wanted to make up for Confucianism. For example, Wang Daiyu (王岱輿) points out the contradictory representations of the relationship between Li and Qi (理氣關係) in different passages of the *Great Book of Xingli* (Wang 1988, pp. 52–53), and he has his own views on traditional Chinese filial piety, arguing that the most unfilial act is to abandon study rather than to have no offspring (Wang 1988, p. 92). Wang Daiyu, Ma Zhu (馬注), and Liu Zhi (劉智) all have their own unique interpretations of the traditional Chinese concepts of "Heaven" and "Mandate". Liu Zhi criticizes Zhu Xi's definition of Heaven and the Principle of Nature (Li) as unclear, and Zhu's logical contradictions in the metaphysical realm raise ethical dilemmas (Liu 1924, pp. 11–13). Starting from the position of the Oneness of Being (wahdat al-wujūd), Liu Zhi argues that what Neo-Confucians call the Principle of Nature or Li is not the absolute Being (God) Himself, but His self-entification (ta'ayyun), i.e., God's Attributes and the essences or entities (a'yān) of all creation, or the Muhammad Reality (haqīqat al-Muhammadī).

As the Qing Dynasty's control in the ideological sphere became stronger since the Qianglong period, Muslim writing gradually fell silent. From the Qianlong period onward, only a few Muslim officials or cultural figures were engaged in writing, and their influence was no longer as great as that of their predecessors. It was not until the end of the Qing period that the writing activities of Islamic scholars were revived to some extent in Yunnan.

**Table 1.** Representatives of Huiru and their important works in the Ming and Qing periods.

| Representatives of Huiru | Identity | Representative Works |
|---|---|---|
| Chen Si 陳思 | Scholar and Leader of Jinan South Mosque | *Inscription of Origin and Return* 來複銘 |
| Zhang Zhong 張中 (1584–1670) | Scholar and Teacher of Jinling School | *Enlightened Explanation of Kalimah al-tayyibah* 克理默解啓蒙淺說<br>*Explanation of Imān al-mujmal* 歸真總義<br>*Translation and Explanation of the Four Essential Chapters* 四篇要道譯解 |
| Wang Daiyu 王岱輿 (ca.1608–1661) | Scholar and Teacher of Jinling School | *Real Commentary on the True Teaching* 正教真詮<br>*Great Learning of Pure and Real* 清真大學<br>*True Answers to Those Who Yearn for Real* 希真正答 |
| Ma Minglong 馬明龍 (1596–1678) | Scholar and Teacher in Wuhan | *Apothegms of Knowing Oneself* 認己醒語 |
| Wu Zunqi 伍遵契 (ca. 1598–1689) | Scholar and Teacher of Jinling and Shandong School | *A Translation of Mirsād al-'ibād* 歸真要道譯義<br>*Guidance for Children to Practice the Truth* 修真蒙引 |
| She Yunshan 舍蘊善 (1638–1703) | Scholar and Teacher of Shandong School | *The True Path of Pursuing the Origin* 推原正達 (A Translation of *Mirsād al-'i bād*)<br>*The Secret of Revealing the Origin* 昭元秘訣 (A Translation of *Ashi"at al-Lam'āt*)<br>*Necessary for Return to the Real* 歸真必要/*The Classic of Searching for the Real* 研真經 (A Translation of *Maqsad al-Aqsā*) |
| Mi Wanji 米萬濟 (1584–1670) | Scholar and Teacher of Shandong School | *A Brief Discussion of Religious Matters* 教款微論 |
| Ma Zhu 馬注 (1640–1711) | Scholar in Yunnan | *Guide to Pure and Real* 清真指南 |

**Table 1.** *Cont.*

| Representatives of Huiru | Identity | Representative Works |
|---|---|---|
| Liu Zhi 劉智 (ca.1660–ca.1730) | Scholar of Jinling School | *Nature and Principle in Islam* 天方性理<br>*Norms and Rites in Islam* 天方典禮<br>*The Biography of the Utmost Sage of Islam* 天方至聖實錄<br>*The Significance of the Five Pillars* 五功釋義<br>*Displaying the Concealment of the Real Realm* 真境昭微<br>*Three Character Classic in Islam* 天方三字經<br>*Verses of the Moon in Five Watches of the Night* 五更月偈 |
| Hei Mingfeng 黑鳴鳳 (1665–1719) | Imperial Bodyguard | *Annotation the Root Classic in Nature and Principle* 性理本經註釋 |
| Jin Tianzhu 金天柱 (ca.1700–1781) | Government Official | *Explaining doubts about Pure and Real* 清真釋疑 |
| Mu Rukui 穆汝奎 | Scholar in Zhili | *Explaining the Meaning of Pure and Real* 清真闡義 |
| Ma Deixin 馬德新 (1794–1874) | Scholar and Funder of Yunnan School | *Gathering the Essentials of Four Canon* 四典要會<br>*The Return of the Whole Creation* 大化總歸<br>*Chinese Translation of All the Details of the Way* 漢譯道行究竟 (A Translation of *Mirsād al-'ibād*)<br>*The Gist of Nature and Mandate* 性命宗旨<br>*The Classic of the Clear Character* 明德經 |
| Lan Xu 藍煦 (1813–1876) | Government Official | *True Learning in Islam* 天方正學<br>*Literary Expositor in Islam* 天方爾雅<br>*Commentary on the Book of Changes* 易經全解 |
| Ma Anli 馬安禮 (1820–1899) | Scholar of Yunnan School | *The Book of Odes in Islam* 天方詩經 |
| Tang Jinhui 唐晉徽 (1820–1900) | Government Official | *A Supplementary Collection of Explaining the Meaning of Pure and Real* 清真釋疑補輯 |
| Yang Jingxiu 楊敬修 (1870–1952) | Scholar and Teacher of Shangdong School | *A Brief Summary of the Four Teachings* 四教要括<br>*Commentary on the Classic of Teaching the Heart* 教心經註 (A Translation of *Sharh al-'Aqā'id al-Nasafiyyah*) |

## 4. Huiru Study of Xingli (Nature and Principle)

The Study of Xingli (性理学), a term derived from Neo-Confucianism, is undoubtedly the most important area of study for Huiru. In fact, the term is used by Huiru to refer to the common concerns of Islam and Confucianism; that is, the transformation of the human self and the achievement of "one body with Heaven and earth" (天人合一). According to Liu Zhi's understanding, Xing or Nature refers to the soul of the microcosm (i.e., man himself) and Li refers to the essence and principle of the macrocosm. The Study of Xingli requires that the external cosmic essence be recognized through the inner nature of man; through man's efforts to cultivate himself (knowing the nature and nurturing the nature), the ultimate realization is the unity of heaven and man. In the view of Huiru, this learning is not only contained in the teachings of Islam, especially Sufism, but also in the teachings of Confucianism—and even Buddhism and Taoism. As Chittick says, "those who study the writings of the Huiru in the context of both Islamic and Chinese thought will find a rich repository of teachings about the human situation in relationship with Heaven, earth, and the ten thousand things. They will discover that many of the obstacles that make it difficult for Muslims to understand the Three Teachings, and for Chinese to understand Islam, disappear in the logical rigor of Huiru thought. They will see that these scholars carried out a fruitful dialogue between Islam and Confucianism, a dialogue that addresses the most important issues of human embodiment and human flourishing. They will gain greater insight into our common humanity, and that is certainly a primary goal of any institute of humanistic studies" (Chittick 2011).

### 4.1. Sufism: Islamic Study of Xingli

Sufism spread to mainland China as early as the Song and Liao periods. The two Shaykhs, Ahmad Burtānī and ʻAlī ʻImād al-Dīn, who were buried in the Niujie Mosque in Beijing, came to China during the Liao period. Bahāʼ al-Dīn, who died in 1275 and was buried in Yangzhou, was a prominent Sufi who came to China during the Song Period. During the Yuan period, Sufis came to China more commonly, and the term Darwīsh (迭里威士) appears frequently in histories of the Yuan Dynasty. According to Ibn Baṭūṭah's Travels, Sufi monasteries were built in Quanzhou, Guangzhou, Beijing, and other cities as places for Sufi practice and preaching. In the late Yuan and early Ming periods, it was said that a Sufi Shaykh named Hamzah led 40 Shaykh Qutb to Hezhou and then spread out to preach in Gansu, Ningxia, Qinghai, Shaanxi, and Sichuan (Li et al. 2007, pp. 201–3). By the time of Hu Dengzhou, Sufism had gained widespread influence in China, and most of the teachers and scholars of early Jingtang education had a close relationship with the Sufis. Chinese Muslim scholars soon discovered that there were many commonalities between Sufism and the Study of Xingli in China. In order to adapt to mainstream discourse, they also called Sufism by this name (Wang 2022).

The most important scriptures that Huiru referred to in their writings were Sufi texts. Sufi books account for a large proportion of both the textbooks of Jingtang education and the bibliography of scholars. Five of the Thirteen Classics, which were compulsory for early Jingtang education, are related to Sufism. They are all Persian texts, namely, *The Sermons of the Messenger* (*Khutab Rasūl*), *The Forty Hadiths of the Poor* (*Arbaʻūn Fuqarāʼ*), *The Roses Garden* (*Gulistān*), *The Path of God's Bondsmen from Origin to Return* (*Mirsād al-ʻibād min al-mabdaʼ ilaʼl-maʻād*), and *The Rays of the Flashes* (*Ashiʻʻat al-lamʻāt*). In *Tianfang Xingli* (天方性理) and *Tianfang Dianli* (天方典禮), Liu Zhi lists a total of 65 reference books, covering more than ten fields, such as commentaries on the Qurʼan, jurisprudence, theology, Sufism, logic, history, astronomy and the calendar, geography, linguistics, and dream interpretation. Scripture related to Sufi thought accounts for nearly a quarter of these books. Of all the Sufi texts referred to by Huiru, four were the most influential. *The Path of God's Bondsmen* is the most famous book of Rāzī, who was a disciple of Najm al-Dīn Kubrā (d. 1221), the eponymous founder of the Kubrawī Order (Murata et al. 2009, p. 11). This book was translated into Chinese as *Guizhen Yaodao* (歸真要道) by Wu Zunqi (伍遵契) and as *Tuiyuan Zhengda* (推原正达) by She Yunshan (舍蘊善). *The Rays* is a commentary on the Persian prose classic *The Flashes* (*Lamaʻāt*) by Fakhr al-Dīn ʻIrāqī (d. 1289), a student of Ṣadr al-Dīn Qūnawī (d. 1274). This book plays an important role in the school of Ibn al-ʻArabī. It was translated into Chinese as *Zhaoyuan Mijue* (昭元秘訣) by She Yunshan in 1679. Jāmī's *Gleams* (*Lawāʼih*) is called Zhaoweijing (昭微經) in Liu Zhi's Tianfang Xingli, where Liu cites it eleven times. He later translated this book into Chinese as *Zhenjing Zhaowei* (真境昭微). ʻAzīz al-Dīn al-Nasafī's *The Furthest Goal* (*Maqsad al-Aqsā*), was translated into Chinese as *Guizhen Biyao* (歸真必要) or *Yanzhenjing* (研真經) by She Yunshan.

The above four Persian Sufism works had a great influence on the history of Chinese Islamic thought and played a key role in the Chinese writing activities during the Ming and Qing periods. In contrast, Huiru adopted limited resources from philosophical and theological texts such as al-Mawāqif (格致全經), Sharh al-Mawāqif (格致經解), and Sharh al-ʻAqāʼid (教典釋難), and seem to have adopted only some doctrines about the Essence and Attributes and ideas of the four elements. In addition, they adopted only the theory of universals related to the five predicates from logical works such as Wazāʼif (六十廩), al-Shamsiyya (明理真宗), and Isāghūjī (明理解).

### 4.2. Resources of Chinese Confucianism, Taoism, and Buddhism

The traditions of Confucianism, Taoism, and Buddhism since the Song, Yuan, and Ming Dynasties are important ideological resources for the Huiru Study of Xingli. Just as Islamic thought inevitably bears the imprint of Greek philosophy in the West due to its collision and integration with Greek philosophy, Islamic thought is bound to have an interactive relationship with the Chinese ideological tradition from the ancient East. Chinese

Muslims in the Yuan and Ming periods lived in a cultural context where Confucianism was the mainstream and the Three Teachings tended to merge, and their understanding of Islamic teachings was deeply influenced by these traditions. During the Song, Yuan, and Ming periods, the convergence of the Three Teachings entered a new stage and reached unprecedented heights. Professor Mu Zhongjian (牟鐘鑒) summarizes its characteristics as the following five points: first, Confucianism created Neo-Confucianism by promoting the body of transmitted orthodox teachings and absorbing Buddhism and Taoism, climbing to a new theoretical peak. Second, while adhering to their own classics and core beliefs, the people of the Three Teachings also studied and grasped the classics and essential principles of the other two teachings, thus bringing the integration of the Three Teachings to a philosophical height. Third, the theories of the Three Teachings reached a high degree of unity in the study of mind and nature (心性之學), building the philosophical foundation of the fusion of the Three Teachings. Fourth, there were many thinkers among the Three Teachings who advocated "the unity of the Three Teachings" and the division of labor and cooperation to maintain social order and human morality, improve human nature, promote good, and suppress evil. Fifth, four classics played a central role in the convergence of the Three Teachings, namely *The Four Books* of Confucianism, the *Lao Zi* of Taoism, the *Book of Changes* of Confucianism and Taoism, and the *Platform Scripture* of Buddhism (Mu 2018, pp. 369–75).

4.2.1. The Tiantai School, the Huayan School, and the Zen (Chan) School

As typical representatives of Chinese Buddhist thought, the ideology of the Tiantai, Huayan, and Zen schools not only influenced Confucianism in the Song and Ming periods but can also be found in the writings of Huiru. During the Sui and Tang periods, Buddhist sects flourished and created theoretical systems distinct from Indian Buddhism by adapting Chinese culture to the Indian Buddhist classics. The idea of Tathāgatagarbha (如來藏) or the Pure Mind of the Self (自性清淨心) in the *Mahāyāna śraddhotpada śāstra* (大乘起信論) had an important founding role in the emergence and development of Chinese Buddhist sects in the Sui and Tang periods. The Huayan School, founded by Fa-tsang (法藏, 643–712), was named after the *Avatamsaka sutra* (華嚴經) and centers around its fundamental concept, the Universal Causation of the Realm of Dharmas (法界緣起). According to Huayan Buddhism, all phenomena in the world and outside the world arise from the pure mind of the Self (the true Dharma Nature). Each dharma is at once one and all and the world is in reality a Perfect Harmony. Consequently, when one dharma rises, all dharmas rise with it, and vice versa. In short, the entire universe rises at the same time. All sentient beings have the wisdom and virtue of the Buddha, but they are unable to realize it because of delusion and attachment. The standard sayings of the Zen School are: "Point directly to the human mind" and "See one's nature and become a Buddha". The Sixth Patriarch of Zen Buddhism, Hui Neng (惠能), advocated the idea that the mind is pure, that the Buddha nature is inherent, and that enlightenment is not sought from outside but is self-realized and self-seen. He said, "therefore, when they meet a good and learned friend who reveals to them the true method and scatters delusions and falsehood, then they are thoroughly illumined both internally and externally, and all dharmas reveal the free and easy character in their own nature. This is called the Pure Law-body" (Chan 1969, p. 438). Zen Buddhism believes that all human beings have the true Buddha Nature and that all dharmas arise from ignorance (無明). The above ideas were used by Muslim scholars to express the theory of "the Muhammadan Reality" in Sufi thought, which is particularly evident in the late Ming Muslim scholar Zhang Zhong's *Guizhen zongyi* (歸真總義). One of the most important ideas that Zhang Zhong wanted to express was "the annihilation of the appearance of self and the present of God" (我相融而真主現). He says "when a man is born, he has the appearance of self. Since they have the appearance of self, they have these three paths under their feet, the middle path and the left and right sides. Therefore, it is said the root of heaven, the seed of hell, but the only true path is to put down the whole body" (Zhang 2005, p. 244). It is said that people should eliminate all kinds of delusions and pursue the realm of "no-self" through

the work of "submissiveness", so that they can finally realize their "original face" (本來面目) and not lose their "original understanding Wisdom" (本明智慧); that is, Rūh al-iḍāfī (继性) in Sufism. In addition, the works of scholars such as Zhang Zhong, Wang Daiyu, and Ma Zhu all contain borrowings of many other Buddhist terms, such as Zhenru (真如), Yuanqi (缘起), Sejie (色界), Miaodi (妙谛), Wuchang (无常), Ji (偈), etc.

### 4.2.2. The Internal Alchemy Taoism

The concept of Taoist internal alchemy (内丹), represented by Quanzhen Taoism (全真道), also became an important resource for the translation and interpretation of classics by Huiru. The Quanzhen Taoism founded by Wang Chongyang (王重陽, 1112–1170) elevated Taoism from a religion that worships ghosts and gods to a religion of the liberation of the body and mind. He changed the previous Taoist emphasis on external ceremony, talismans, spells, alchemy, and other magical arts to the transcendent concept of the inner elixir of life and well-being. This new Taoism, called the Internal Alchemy, emphasizes true pure and true tranquility (真清真靜), holding the origin and keeping the One (抱元守一), and preserving one's spirit and fixing one's Qi (存神固氣). The eternal life understood by the Internal Alchemy Taoism is not the physical ascension of their predecessor Ge Hong(葛洪) but the eternal residence of the spirit; "the true nature is not disturbed, and all karmas are not attached, not going and not coming, this is the immortality of eternal life" (Mu 2018, p. 342). They were guided by the idea of the unity of man and nature, treating the human body as an alchemical furnace and adopting the practice of internal refinement of spirit and Qi. According to the Internal Alchemy Taoism, man is a microcosm and his path of cultivation should be to reverse the macrocosmic evolutionary pattern of "Tao produced the One, the One produced the two, the two produced the three, and the three produced the ten thousand things"; that is, from many to the two and from the two to the One. This idea of "Heaven obtained the One and became clear; Earth obtained the One and became tranquil; The spiritual beings obtained the One and became divine" and "return to the Root and the Mandate", which was proposed by Laozi (老子), is most clearly reflected in the Huiru Ma Minglong's book *Apothegms of Knowing Oneself* (認己醒語). In this book, Ma explains the Dao (tarīqah) of Sufism and the theory of *Oneness of Being* using Taoist terms such as the absolute beginning (太初), the absolute pure (太清), wuxiang (無相), the true One (真一), the door of all subtleties (眾妙之門), the original Nature (元性), and concepts of the Internal Alchemy Taoism (Wang 2021). This tendency of Ma Minglong to "explain the scriptures by the Tao" also influenced the later scholar Liu Zhi to some extent.

### 4.2.3. Neo-Confucianism

Among the Three Teachings, the most important intellectual resource for Huiru is the Confucian tradition, especially Neo-Confucianism. Confucianism in the Song and Ming periods changed its passive status from the Wei and Jin (魏晉) to the Sui and Tang (隋唐) periods, when it could only hold onto political and moral positions, while lagging behind Buddhism and Quanzhen Taoism in philosophy, and regained its dominant position among the Three Teachings, with Buddhism and Taoism as its supporting wings (Mu 2018, p. 370). Zhou Dunyi (周敦頤, 1017–1073) relied on the *Book of Changes* and the *Doctrine of the Mean* and wrote *An Explanation of the Diagram of the Great Ultimate* (太極圖説) and *Penetrating the Book of Changes* (通書), in which he laid the pattern of metaphysics and ethics for later Neo-Confucianism. He was a pioneer of Neo-Confucianism who was dedicated to the continuation of authentic Confucianism. Zhang Zai (張載, 1020–1077) lived at roughly the same time as Zhou Dunyi. His ideas such as the Great Vacuity (太虛), the original nature endowed by Heaven and Earth (天地之性) and physical nature (氣質之性), and his concept of Heaven and Earth as universal parents and love for all had a profound influence on later generations. After Zhang Zai, the two Cheng (二程) brothers established the core position of the principle (Li) and set the tone for the different directions of Confucianism afterwards. In the Southern Song period, Zhu Xi (1130–1200) set up a huge ideological system by integrating the great achievements of Confucianism, and his philosophy became

the official school of the Yuan and Ming Dynasties. In the middle and late Ming periods, Wang Yangming's Study of Mind emerged, which was dedicated to breaking the authority of traditional thought, correcting the shortcomings of Zhu's Study, and opening up a new atmosphere, which, as pointed out earlier, had an important influence on the rise of Chinese Islamic scholarship. In short, the ontology, cosmology, psychology, and ethics established by Neo-Confucian scholars became important intellectual resources for the writings of Huiru. The concepts and terminology of essence-function (), the Ultimate of Non-being (無極) and the Great Ultimate (太極), Yinyang (陰陽) and the Five Agents (五行), Principle (理) and material force (氣), mind and nature (心性), the Five Relationships (五倫), the Eight Virtues (八德), the unity of heaven and man, or one body with heaven and earth are common in the writings of Huiru. Huiru were in an academic environment dominated by the concepts of Dao, Nature, Mandate, Principle, Qi, Heart, and Mind, so they used the same words and concepts in their writings to expound Islamic thought and to seek integration and common understanding between Chinese and Islamic civilizations.

## 5. Conclusions

Different scholars have come to different conclusions about the causes for the production of Islamic writings in Chinese during the Ming and Qing periods. Some scholars describe it as the "cultural self-awareness" of Chinese Muslim scholars (Kuwata 1925); others believe that their original intention of writing books was to preach Islam and pass on Muslim culture as well as to gain the understanding and sympathy of non-Muslims in China (Ma 2014, p. 11). Some scholars believe that the more developed socio-economic conditions of the Jiangnan area during the Ming period led to a vibrant atmosphere in the field of thought (Qiu 1996, p. 527). Some scholars emphasize that the Jesuits' missionary writings attached to Confucianism and their successful preaching to Confucian scholars in China stimulated the rise of Muslim writing activities (Ma 2022). Some scholars, in order to prove that Chinese Muslim writing activities in the Ming and Qing periods were a direct product of the stimulation of Catholic writing in Chinese, have gone to the extent of pitting Jingtang education against the writing activities, arguing that Chinese Muslim writing activities emerged in areas where Jingtang education was "underdeveloped" (Wen 2007). In fact, Chinese Muslim scholars benefited from Jingtang education, and the more developed the Jingtang education was in the early days, the more their writing activity flourished. It is clearly untenable to separate Jingtang education from their writing or even to consider the two to be mutually exclusive. In any case, we do not deny any of the factors mentioned above, but neither do we completely fall back on any single position. As stated at the beginning of this paper, the emergence of any kind of thought or cultural phenomenon is closely related to a specific social context, and to some extent, it is inevitably influenced by a combination of factors such as the social group, cultural system, historical context, and spirit of the times. If one of these factors is overemphasized to the detriment of the others, the conclusion reached will inevitably be biased.

**Funding:** This research is a phased result of the Youth Project of the National Social Science Fund of China (Grant No. 21CZJ018).

**Institutional Review Board Statement:** Not applicable.

**Informed Consent Statement:** Not applicable.

**Data Availability Statement:** Not applicable.

**Conflicts of Interest:** The author declares no conflict of interest.

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
