# Peer review of "On the Historical Background and Ideological Resources of the Confluence of Islam and Confucianism"

_religions, doi:10.3390/rel13080748_

Round 1

Reviewer 1 Report

The article examines the interaction of Muslims with Confucianism (which also touches upon interaction with Chinese Buddhism and Taoism at the end of the article) during different Chinese dynasties, chronologically. The article, which draws attention to the inscriptions in the mosques of different periods, summarizes well the effort of Muslims trying to find a place in Chinese society. Although the author draw attention to the tension experienced from time to time, Islamic dogmas in accordance with the socio-cultural and political environment in Chinese languages are explained convincingly. Generally speaking, this is a successful work with its rich references and logical argument. It would be very useful to publish it in a not very well-known area.

Author Response

Dear reviewer,
Thank you for your comments concerning my manuscript. I will make appropriate English revisions to my manuscript, and continue to study this topic in depth.

Reviewer 2 Report

This essay makes useful contributions in two important fields. One is the primary and explicitly stated subject of the essay -- the history of Muslim-Confucian relations in China. However, in addition, the analysis makes an important contribution to the study of trans-religious relations and interreligious dialogue. This aspect of the study  is not as explicitly defined by the author but it provides a good case study of interreligious relations that goes beyond the old-style "world religions" paradigm. The "world religions" approach views "religions" as clearly defined and separate entities which may "borrow" from each other, but remain separate, while this essay shows clearly the synergetic fluidity of religious discursive traditions.

The translations of mosque inscriptions and the textual analysis by the essay's author provide a very important set of sources fir understanding Muslim-Confucian relations. The inscriptions were written by Confucian scholars and provide an important source for understanding how they understood Islam in Confucian terms. This analysis is an important contribution to the scholarship since, as the author notes, much of the scholarship examines the thinking of Muslim scholars and gives less attention to the work of Confucian scholars.

In this analysis and in the essay as a whole, the author uses the term "Confucianism." There is a long debate over terminology which notes that "Confucianism" is a term originally applied by non-Chinese observers studying Chinese history and culture. It might be helpful if the author could provide a short note on why the term is used in this study.

Another important part of this study is the discussion of Sufism and Chinese religious tradition. The author notes the importance of Sufism in religious interrelationships in China. The analysis shows how these relationships were influenced by the evolution of Confucian thought and the history of neo-Confucianism. However, the author tends to present Sufism as a fixed tradition. The analysis could be strengthened by noting the evolution of Sufism -- the Sufism that came to China during the Song was quite different in terms of institutional organization and theology from what it became by the Qing period. This changing context helped to shape the evolving nature of "the confluence of Islam and Confucianism" and it would be helpful if it received a bit more attention in this analysis.

In the discussion of Muslim scholars like Wang Daiyu (see page 9), the author might find it useful to look at the study of Wang Daiyu's writings by Lee Cheuk Yin in the volume edited by Osman Bakar, Islam and Confucianism: A Civilizational Dialogue (ISTAC-IIUM Publications [Malaysia], 20190.

An important theme in this study is the analysis of how Islam was integrated into Chinese tradition(s), what the author calls the "Sinicization" of Muslims and Islamic tradition. The study of the successful localization of Islamic tradition in different societies and cultures in an important subject in global Islamic studies. This essay will be a very useful addition to this scholarship, showing the importance of what some scholars have called the "glocalization" of religion in world history.

Author Response

Dear Reviewer,

Thank you for some very good suggestions on my manuscript. Your comments and suggestions are very helpful for the improvement of my article.